# Neurofeedback training can modulate task-relevant memory replay rate in rats

Anna K Gillespie[1,2]*, Daniela Astudillo Maya[2], Eric L Denovellis[2,3], Sachi Desse[2], Loren M Frank[2,3]

[1]Departments of Biological Structure and Lab Medicine & Pathology, University of Washington, Seattle, United States; [2]Departments of Physiology and Psychiatry and the Kavli Institute for Fundamental Neuroscience, University of California, San Francisco, San Francisco, United States; [3]Howard Hughes Medical Institute, Chevy Chase, United States

**Abstract** Hippocampal replay – the time-compressed, sequential reactivation of ensembles of neurons related to past experience – is a key neural mechanism of memory consolidation. Replay typically coincides with a characteristic pattern of local field potential activity, the sharp-wave ripple (SWR). Reduced SWR rates are associated with cognitive impairment in multiple models of neurodegenerative disease, suggesting that a clinically viable intervention to promote SWRs and replay would prove beneficial. We therefore developed a neurofeedback paradigm for rat subjects in which SWR detection triggered rapid positive feedback in the context of a memory-dependent task. This training protocol increased the prevalence of task-relevant replay during the targeted neurofeedback period by changing the temporal dynamics of SWR occurrence. This increase was also associated with neural and behavioral forms of compensation after the targeted period. These findings reveal short-timescale regulation of SWR generation and demonstrate that neurofeedback is an effective strategy for modulating hippocampal replay.

*For correspondence:
annagill@uw.edu

## eLife assessment

This study tests the effects of using neurofeedback, in the form of reward delivery when large sharp wave-ripples (SWRs) are detected, on neurophysiological and behavioral measures. The results are **important**, and the authors provide **convincing** evidence that the rate of SWRs increased prior to reward delivery and decreased in the period after reward delivery, with no significant effect on memory performance. The ability to manipulate SWR rate in a naturalistic way is an exciting new tool for studies that seek to understand the function of SWRs.

## Introduction

Across species, memory is central to adaptive behavior. Key memory processes are coordinated by the hippocampus, where highly plastic neural ensembles are thought to rapidly encode ongoing experience. Subsequently, patterns of neural activity associated with past experience are 'replayed' during offline states, such as sleep and pauses in ongoing behavior (*Findlay et al., 2020*). Replay events tend to coincide with sharp wave ripples (SWRs), characteristic bursts of high-frequency oscillatory activity in the CA1 region of the hippocampus that coordinate spiking activity in a wide range of cortical and subcortical brain regions (*Buzsáki, 2015*; *Joo and Frank, 2018*; *Skelin et al., 2019*; *Todorova and Zugaro, 2020*). The brain-wide coordination of activity during SWRs is thought to enable their proposed role in storing and maintaining memory traces in distributed hippocampal-cortical circuits (*Frankland and Bontempi, 2005*; *Kaefer et al., 2022*; *Klinzing et al., 2019*).

Disrupting SWRs impairs learning and alters memory performance, establishing the necessity of SWRs for intact memory-guided behavior (*Aleman-Zapata et al., 2022*; *Ego-Stengel and Wilson, 2010*; *Girardeau et al., 2009*; *Jadhav et al., 2012*; *Nokia et al., 2012*; *Todorova and Zugaro, 2020*). The importance of SWRs to memory function is further underscored by observations of reductions and abnormalities in SWRs concurrent with memory impairment in several rodent models of neurological disease (*Gillespie et al., 2016*; *Iaccarino et al., 2016*; *Jones et al., 2019*; *Prince et al., 2021*; *Sanchez-Aguilera and Quintanilla, 2021*; *Stoiljkovic et al., 2018*). These findings raise the compelling possibility that driving or enhancing SWRs in cases where they are compromised could confer therapeutic benefit.

Indeed, a small number of previous studies have demonstrated positive manipulations of SWRs. Electrical stimulation and optogenetics have been used to evoke exogenous SWRs (*Ishikawa et al., 2014*; *Jiang et al., 2018*; *Stark et al., 2015*; *Stark et al., 2014*, *Oliva et al., 2020*) or to enhance existing SWRs (*Fernández-Ruiz et al., 2019*; *Maingret et al., 2016*). While compelling, some limitations remain. In some cases, studies have demonstrated the selective engagement of subsets of hippocampal neurons (*Fernández-Ruiz et al., 2019*; *Oliva et al., 2020*), but whether these manipulations preserve the full diversity or accuracy of naturally occurring replay events remains unknown. Further, endogenous SWRs tend to occur within permissive windows influenced by cortical activity states (*Battaglia et al., 2004*; *Isomura et al., 2006*) and neuromodulatory tone (*Novitskaya et al., 2016*; *Vandecasteele et al., 2014*); it is unclear whether exogenous events evoked outside of such states recruit downstream targets or engage brain-wide coordination as endogenous SWRs do. Finally, both electrical and optogenetic interventions are suboptimal for translation to human use. Both approaches would require invasive procedures, and clinical use of optogenetics in the brain faces even more fundamental barriers (*White et al., 2020*).

These limitations motivated us to take an alternative approach. Neurofeedback, wherein subjects are given real-time external feedback that allows them to alter their own brain states, has been shown to enable modulation of patterns of brain activity across species (*Sitaram et al., 2017*). The efficacy of neurofeedback-based therapy has been established for some clinical indications, such as ADHD and stroke, but the use of neurofeedback for memory-related symptoms has shown variable efficacy (*Staufenbiel et al., 2014*; *Wang and Hsieh, 2013*) and is still in early stages (*Jiang et al., 2022*; *Klink et al., 2021*; *Trambaiolli et al., 2021*). Moreover, most neurofeedback paradigms target the modulation of oscillatory activity in specific frequency bands linked to cognitive functions, such as gamma or theta (*Engelhard et al., 2013*; *Reiner et al., 2014*; *Wang and Hsieh, 2013*) or overall activity levels (*Klink et al., 2021*), rather than focusing on specific memory-related hippocampal activity patterns (e.g. SWRs).

We therefore asked whether providing external positive neurofeedback coupled to SWRs would effectively drive SWR occurrence, a possibility consistent with prior work using intracranial stimulation of a reward-related brain region to achieve operant conditioning of SWRs (*Ishikawa et al., 2014*). Specifically, we set out to determine whether subjects could learn to use noninvasive positive neurofeedback to achieve a SWR-conducive state and increase SWR generation. Further, since the replay during SWRs is thought to be central to their function, we also sought to evaluate whether SWR-based neurofeedback could drive replay events with physiologically relevant spatial content.

We were able to develop a neurofeedback training paradigm for rats that increases the rates of SWRs and replay during the targeted neurofeedback period. Importantly, SWRs during the targeted period were reliably associated with replay events which represented a range of spatial content consistent with that seen in our previous study of replay in a very similar task (*Gillespie et al., 2021*). Interestingly, we also observed some compensatory changes outside of the targeted neurofeedback period. Our results demonstrate the efficacy of a neurofeedback strategy to modulate hippocampal replay in young, healthy animals. Further, they suggest that SWR generation during task performance is influenced by homeostatic regulation on the timescale of seconds, an important factor to consider when developing strategies to drive SWRs and replay therapeutically.

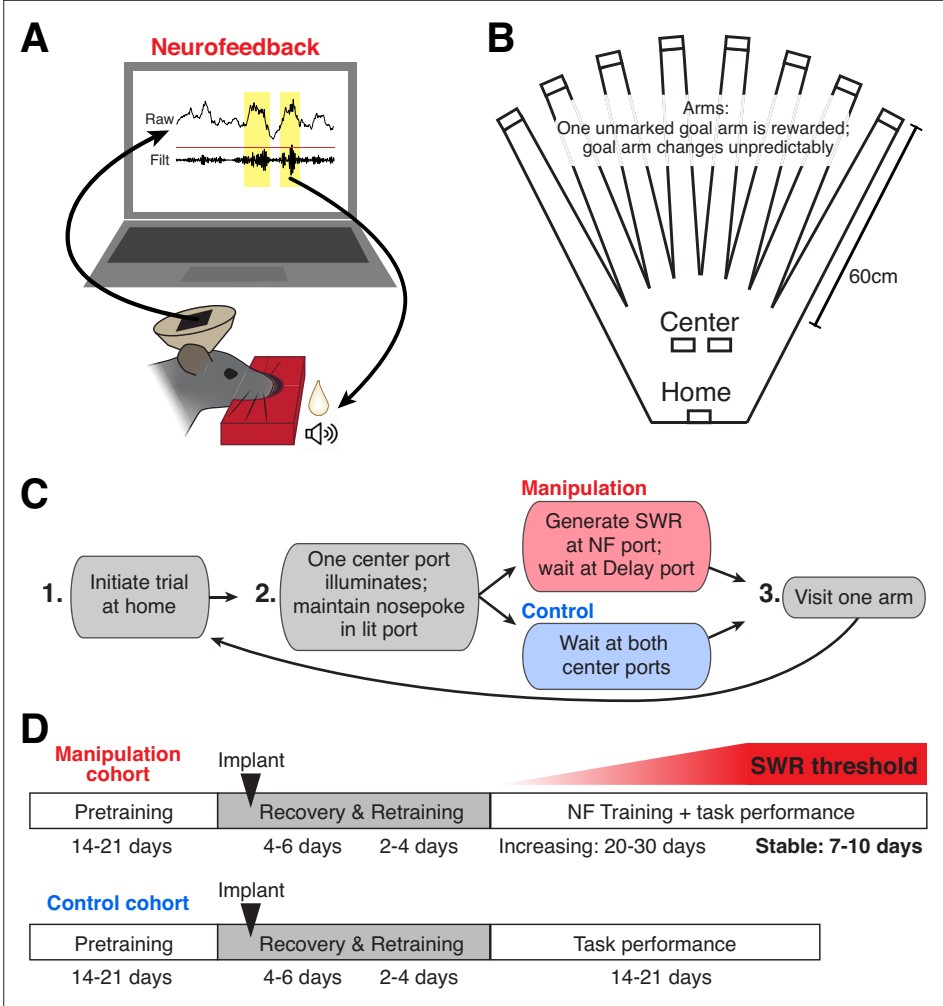

**Figure 1.** A neurofeedback paradigm to promote SWRs. (**A**) Schematic of the neurofeedback (NF) protocol: while subject's nose remains in the neurofeedback port, SWRs (yellow) are detected in real time. Top trace: raw CA1 LFP; bottom trace: ripple filtered (150–250 Hz) CA1 LFP. During the neurofeedback interval, the first event exceeding a set standard deviation (sd) threshold (red line) triggers the delivery of a sound cue and food reward to the rat. (**B**) Top-down view of the maze environment. Reward ports are indicated by rectangles. (**C**) The rules of the spatial memory task for each of the behavioral cohorts. (**D**) Experimental timeline for each behavioral cohort.

The online version of this article includes the following figure supplement(s) for figure 1:

**Figure supplement 1.** A neurofeedback paradigm to promote SWRs.

## Results

### A neurofeedback paradigm to promote SWRs

To reinforce SWR activity, our neurofeedback paradigm linked the online detection of large SWRs with positive reinforcement: delivery of a tone and food reward (*Figure 1A*). We embedded the neuro-feedback requirement into a flexible spatial memory task (*Figure 1B*) that we have studied exten-sively (*Gillespie et al., 2021*), allowing us to compare neural and behavioral data from subjects that perform the task with neurofeedback manipulation to those without the manipulation component. All subjects performed the same overall task, which comprised structured trials consisting of three steps (*Figure 1C*): first, initiating a trial by nosepoking at the home port; second, maintaining a nosepoke at whichever center port illuminates (randomly assigned on each trial) for a duration determined by the port's criteria (see below); and finally, choosing one of eight arms to visit before returning to the home port to initiate the next trial.

Here we present data from two cohorts of animals. In the manipulation cohort, one center port served as the neurofeedback port and the other served as the delay port. At the neurofeedback port, the subject had to maintain its nose in the port until a suprathreshold SWR was detected, which triggered delivery of a tone and food reward (*Figure 1A*, *Figure 1—figure supplement 1A and B*; and see Methods). The pre-reward period at the center port on neurofeedback trials is thus considered the 'targeted interval' for neurofeedback. At the delay port, the subject was required to maintain its nose in the port for an unpredictable amount of time. The delay period length was randomly chosen from the pre-reward duration of recent neurofeedback trials, such that the amount of time spent waiting at each of the center ports was approximately matched (*Figure 1—figure supplement 1C*; and see Methods). Delay trials thus serve as an internal control for each subject, allowing us to assess the specificity of any neurofeedback effect. In the control cohort, whose data have been presented previously (*Gillespie et al., 2021*), both center ports required the subject to maintain its nose in the port for an unpredictable delay period (see Methods), after which a tone and food reward was delivered.

The memory aspect of this task is required at the third step of each trial: at any given time, one of the eight arms provides a large food reward; the others provide nothing. Over multiple trials, the subject must sample arms to discover which one provides reward and remember to return to that location on subsequent trials. Occasionally, the rewarded arm changes, prompting the subject to return to sampling different arms to find the new goal location (*Gillespie et al., 2021*). Before implant, all subjects were pretrained on a delay-only version of this task, as summarized in the experimental timelines in *Figure 1D*. Once performance stabilized, a microdrive containing 30 independently movable tetrodes was implanted targeted to dorsal CA1 (*Figure 1—figure supplement 1D*). After implant, subjects were given several days to recover; during this time tetrodes were gradually lowered into the CA1 cell layer and the animal was briefly re-trained to perform the delay-only version of the task. The control cohort of subjects continued such performance for many days (*Gillespie et al., 2021*).

For subjects in the manipulation cohort, the neurofeedback component of the task was then introduced. Each day, 4–6 CA1 tetrodes were used for online SWR detection (*Figure 1—figure supplement 1A*; and see Methods). Because we use just a small subset of tetrodes, our online detection algorithm does not detect every SWR event identified by our offline detection strategy, which incorporates all CA1 tetrode data. This strategy provides two important benefits. First, it means that rewards are provided to SWR events on an unpredictable, variable schedule – an operant conditioning regime which drives robust, persistent task engagement (*Ferster and Skinner, 1957*). Second, we choose a different subset of tetrodes each day (see Methods) to ensure that subjects cannot learn to trigger the algorithm and earn reward by repeatedly producing a single firing pattern corresponding to specific units on those tetrodes. Initially, the threshold for online SWR detection is set to a low value, requiring only small SWRs to trigger reward. SWR size, as defined by the peak amplitude of the event, can be generally described by a long-tailed distribution, with larger events occurring less frequently than smaller events (*Yu et al., 2017*). Starting with a low threshold ensures that the animals received feedback relatively rapidly and remained engaged early in training.

Over subsequent days of training, the size threshold for SWR detection at the neurofeedback port was gradually increased, requiring the subject to generate larger and larger SWR events on neurofeedback trials in order to proceed with the task (*Figure 1—figure supplement 1E*). We then maintained the threshold at a high level for several additional days of behavior ('stable period').

## Neurofeedback training increases SWR rate during the targeted interval

In response to the increasing SWR detection threshold, subjects in the manipulation cohort successfully produced increasingly large trigger events at the neurofeedback port over the course of training (*Figure 2—figure supplement 1A*). This result confirms that our feedback criteria were enforced as designed and that subjects were capable of meeting the criteria. As the detection threshold was raised, subjects could have met the neurofeedback criteria simply by waiting much longer for increasingly rare suprathreshold events. This was not the case: instead, subjects produced large SWR events much faster than would have been predicted based on the initial prevalence of similarly sized events (*Figure 2—figure supplement 1B*). This suggests that subjects learned a strategy for generating large events that allowed them to meet the neurofeedback criteria more efficiently.

We reasoned that two candidate strategies could have enabled subjects to limit their wait times: learning to increase the average size of SWR events or learning to increase the rate of SWR event generation across all sizes. To evaluate these possibilities, we quantified differences both between the two cohorts and between trial types of the neurofeedback cohort. When evaluating the significance of differences between the neurofeedback and delay trial types *within* each manipulation cohort subject, we used ranksum tests, since this approach allows us to capitalize on the large numbers of measurements collected from each rat without assuming that they were normally distributed. To quantify systematic group effects of cohort, we used linear mixed-effects models (see Methods), since this approach allows us to appropriately account for correlations between measurements taken from each subject (*Yu et al., 2022*).

First, to assess whether subjects learned to directly increase the size of SWR events, we quantified the mean size of SWRs that occured pre-reward at the center ports on neurofeedback and delay trials during behavioral epochs from the stable period. Importantly, to account for the structure of the task, we excluded suprathreshold events detected by our online algorithm during both neurofeedback and delay trials (see Methods). These analyses revealed that subjects did not consistently increase SWR size: only two out of four subjects showed very slight but significant increases (*Figure 2—figure supplement 1C*). We also found no difference in SWR size between either trial type in the manipulation cohort and the control cohort (*Figure 2—figure supplement 1C*, inset). Instead, when we quantified the number of events of each size per behavioral epoch, we found that within the manipulation cohort, neurofeedback trials contained roughly twice as many events relative to delay trials for all SWR sizes (*Figure 2—figure supplement 1D*). These results rule out our first candidate strategy and indicate that subjects met the neurofeedback criteria efficiently by modulating SWR rate, not event size.

The increase in SWR rate pre-reward was associated with striking differences in when SWRs occurred both within and across cohorts that were evident even in individual trials (*Figure 2A*). To assess how SWR rate was changing during the time spent at the center ports, we calculated SWR rate in 0.5 s bins throughout the time subjects spent nosepoked at the center ports, aligning each trial to reward delivery (*Figure 2B*). We excluded the trigger SWR of neurofeedback trials from analysis, since it is guaranteed to occur just prior to reward delivery in each neurofeedback trial and causes a large spike in SWR rate. Subjects in the manipulation cohort all showed high, stable SWR rates before reward delivery, with neurofeedback trials showing higher SWR rate than delay trials. By contrast, subjects in the control cohort rarely generated SWRs during the pre-reward period but showed much higher SWR rates than the manipulation cohort after reward delivery.

To quantify these effects, we calculated the mean SWR rate over each portion of the time at the center ports (*Figure 2C*): the pre-reward period (left), the post-reward period (middle), and the total port period (pre +post; right). Indeed, we found that during the pre-reward period, SWR rates were approximately twice as high on neurofeedback compared to delay trials, and both trial types from the manipulation cohort had much higher rates than trials from the control cohort (*Figure 2C*, left). This effect was not dependent on the trigger SWR, as excluding suprathreshold events from both neurofeedback and delay trials did not change the result (*Figure 2—figure supplement 1E*). This finding indicates that the neurofeedback training caused far more SWRs to occur prior to reward delivery at both center ports in the manipulation cohort, with the most pronounced effect occurring on neurofeedback trials.

These patterns differed after the reward at the center ports, where SWR rate was slightly but significantly higher during delay trials compared to neurofeedback trials in the manipulation cohort subjects but was overall indistinguishable between the cohorts (*Figure 2C*, middle). Relative to the pre-reward period, SWR rates were maintained at a high level during neurofeedback trials, while they increased substantially during delay and control trials (*Figure 2—figure supplement 1F*). Note that the slightly lower SWR rate post-reward on neurofeedback trials compared to delay trials could not be explained by a refractory period following the trigger SWR (*Figure 2—figure supplement 1G*). When considering both pre- and post-reward periods together, the neurofeedback trials still showed higher SWR rates than delay trials within the manipulation cohort, but the differences between each trial type and the control cohort were not significant (*Figure 2C*, right). This surprising result suggested that the pre-reward effects on SWR rate in the manipulation cohort were followed by some compensatory effects post-reward that resulted in a net normalization of total SWR rate across cohorts.

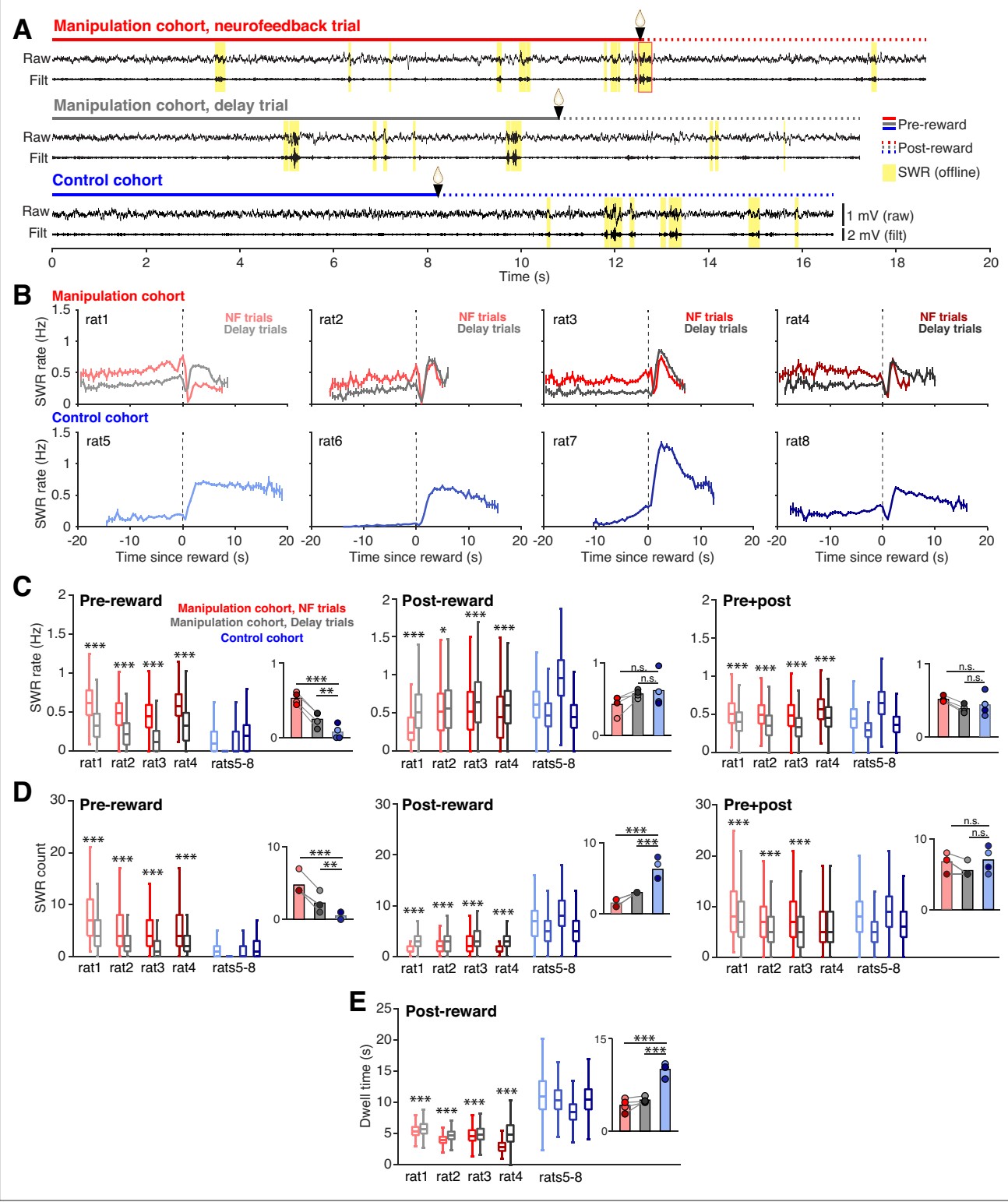

**Figure 2.** Neurofeedback training enhances SWR rate during targeted interval. (**A**) Example CA1 raw LFP traces and ripple filtered LFP traces (150–250 Hz) with SWRs highlighted in yellow, from the time spent at a center port on a neurofeedback (NF) trial (top) and a delay trial (middle) from a manipulation subject and a trial from a control subject (bottom). (**B**) SWR rate calculated in 0.5 s bins during the pre- and post-reward periods, aligned to the time of reward delivery (dashed lines) for subjects in the manipulation (top row) and control (bottom row) cohorts. Trigger SWRs on NF trials are excluded from the rate calculation, and time bins with fewer than 100 trials contributing data are not shown. Vertical bars indicate S.E.M. (**C**) SWR rate calculated during the pre-reward period (left), post-reward period (middle) and for the total time at the center port (right). Manipulation cohort n=1892,

*Figure 2 continued on next page*

*Figure 2 continued*

684, 1157, and 1602 NF trials and 2022, 640, 1201, and 1552 delay trials; control cohort n=2490, 2629, 2027, and 3021 trials. For the pre-reward period, manipulation cohort ranksum comparisons between NF and delay trials: p=4.382 × 10$^{-258}$, 7.111x10$^{-83}$, 5.689x10$^{-214}$, and 3.285x10$^{-191}$. Inset: Groupwise comparisons. Manipulation cohort NF trials vs control cohort trials: p=1.126 × 10$^{-16}$; manipulation cohort delay trials vs control cohort trials: p=0.009. For the post-reward period, manipulation cohort ranksum comparisons between NF and delay trials: p=3.646 × 10$^{-127}$, 0.038, 6.538x10$^{-11}$, and 2.768x10$^{-23}$, respectively. Inset: Groupwise comparisons. Manipulation cohort NF trials vs control cohort trials: p=0.142; manipulation cohort delay trials vs control cohort trials p=0.691. For pre +post combined, manipulation cohort ranksum comparisons between NF and delay trials p=3.324 × 10$^{-64}$, 3.136x10$^{-40}$, 2.996x10$^{-69}$, and 9.066x10$^{-50}$, respectively. Inset: Groupwise comparisons. Manipulation cohort NF trials vs control cohort trials: p=0.237; manipulation cohort delay trials vs control cohort trials: p=0.504. (**D**) Count of SWR events detected during the pre-reward period (left), post-reward period (middle) and for the total time at the center ports (right). Trial n are the same as in (**C**). For the pre-reward period, manipulation cohort ranksum comparisons between NF and delay trials: p=3.133 × 10$^{-96}$, 8.936x10$^{-37}$, 1.266x10$^{-106}$, and 6.024x10$^{-65}$. Inset: Groupwise comparisons. Manipulation cohort NF trials vs control cohort trials: p=6.790 × 10$^{-7}$; manipulation cohort delay trials vs control cohort trials: p=0.0018. For the post-reward period, manipulation cohort ranksum comparisons between NF and delay trials: p=1.797 × 10$^{-137}$, 6.100x10$^{-10}$, 6.688x10$^{-13}$, and 2.152x10$^{-130}$. Inset: Groupwise comparisons. Manipulation cohort NF trials vs control cohort trials: p=2.361 × 10$^{-12}$; manipulation cohort delay trials vs control cohort trials p=5.820 × 10$^{-9}$. For pre +post combined, manipulation cohort ranksum comparisons between NF and delay trials: p=1.534 × 10$^{-19}$, 5.828x10$^{-14}$, 6.735x10$^{-32}$, and 0.083. Inset: Groupwise comparisons. Manipulation cohort NF trials vs control cohort trials p=0.366; manipulation cohort delay trials vs control cohort trials: p=0.299. (**E**), Dwell time post-reward. Trial n are the same as in (**C**). Manipulation cohort ranksum comparisons between NF and delay trials: p=1.397 × 10$^{-23}$, 8.588x10$^{-49}$, 6.180x10$^{-4}$, and 7.127x10$^{-257}$. Inset: Groupwise comparisons. Manipulation cohort NF trials vs control cohort trials: p=4.075 × 10$^{-18}$; manipulation cohort delay trials vs control cohort trials: p=1.835 × 10$^{-18}$. For C-E, all within-subject ranksum p-values are corrected using the Benjamini-Hochberg method and all groupwise comparisons are performed using linear mixed effects models (see Methods).

The online version of this article includes the following figure supplement(s) for figure 2:

**Figure supplement 1.** Neurofeedback training enhances SWR rate during targeted interval.

Quantifying SWR count, rather than rate, provided further explanation for these surprising findings. We quantified the number of SWRs during pre-reward, post-reward, and total center port times (*Figure 2D*). Our SWR count results were completely consistent with the SWR rate findings during the pre-reward period (*Figure 2D*, left) and largely consistent when considering the total center port time (*Figure 2D*, right). However, we found striking differences during the post-reward period, during which we saw a clear reversal of the pre-reward effect on SWR counts: neurofeedback trials contained fewer SWRs than delay trials, and both trial types contained far fewer SWRs post-reward than trials from the control cohort (*Figure 2D*, center). Since post-reward SWR rates were not different between the cohorts, the count results indicated that the compensatory effect must involve the amount of time spent at the port post-reward.

Indeed, we found clear differences in the amount of time that subjects chose to spend at the port after reward delivery. Here it is important to remember that in contrast to the pre-reward dwell time, which is imposed by task constraints, the post-reward dwell time is not externally constrained. It was notable, therefore, that neurofeedback trials had slightly but significantly shorter post-reward periods than delay trials (*Figure 2E*). Even more strikingly, subjects in the manipulation cohort remained at the port for approximately half as long as subjects in the control cohort (*Figure 2E*, inset), despite all trial types receiving the same amount of reward. The difference in dwell time could not be explained by the control cohort having fewer days of task experience than the manipulation cohort, since post-reward dwell time tended to increase, rather than decrease, over subsequent days of performance within the control cohort (*Figure 2—figure supplement 1H*). Together, these findings suggest that both covert and overt processes regulate SWR generation. Animals learned to modulate their SWR rate when required during the pre-reward period of neurofeedback trials and did so covertly, without displaying obvious behavioral correlates. A subsequent overt change in behavior – leaving the well early – was then associated with lower numbers of SWRs post-reward in the neurofeedback cohort.

Of course, the subjects' behavior during the time spent at the ports could have substantial impact on SWR generation processes. In particular, since SWRs are associated with periods of immobility, we wanted to ensure that the increase in SWRs prior to reward delivery during neurofeedback trials was not driven by differences in movement during this time. In general, the requirement for subjects to remain nosepoked forces subjects to remain in the same position; however, small head movements are possible. The mean speeds of the head-mounted LEDs while the subjects were nosepoked were below our cutoff for immobility (4 cm/s) for all trial types, however, we did observe a slightly but significantly lower mean speed during the pre-reward period on neurofeedback trials compared to delay trials and trials from the control cohort (*Figure 3A*). To assess whether this difference could

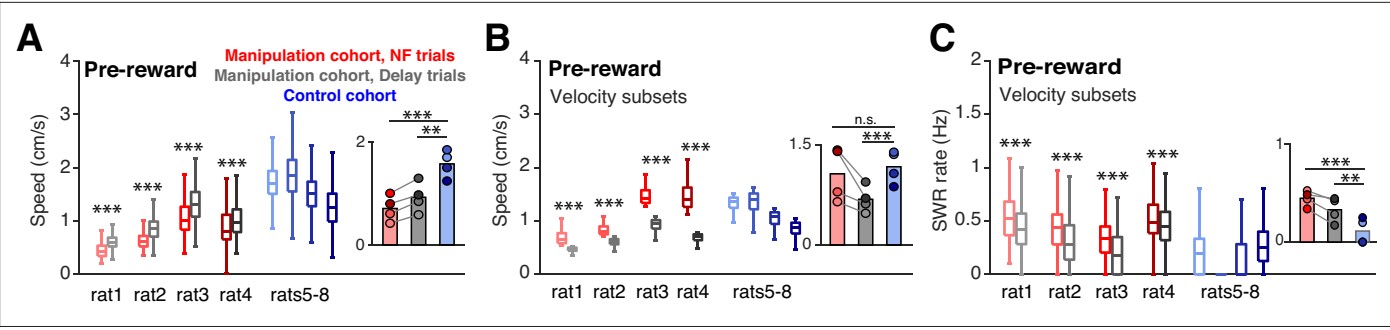

**Figure 3.** Speed does not account for differences in SWR rate. (**A**) Mean head velocity (smoothed), during pre-reward time at the center ports. Trial n are the same as in (**E**); manipulation cohort ranksum comparisons between NF and delay trials: p=2.447 × 10^{-223}, 2.760x10^{-85}, 3.247x10^{-80}, and 9.077x10^{-36}. Inset: Groupwise comparisons. Manipulation cohort NF trials vs control cohort trials: p=1.260 × 10^{-7}; manipulation cohort delay trials vs control cohort trials: p=1.121 × 10^{-4}. (**B**) Mean head velocity for subsets of trials: the quartile of trials with the lowest mean head velocities are shown for the control cohort and for delay trials for the manipulation cohort. The quartile of trials with the highest mean head velocity are shown for neurofeedback trials. Manipulation cohort n=473, 171, 289, and 400 NF trials and 505, 160, 300, and 388 delay trials; control cohort n=624, 794, 423, and 913 trials. Manipulation cohort ranksum comparisons between NF and delay trials: p=1.342 × 10^{-160}, 1.053x10^{-55}, 9.488x10^{-98}, and 4.751x10^{-130}. Inset: Groupwise comparisons. Manipulation cohort NF trials vs control cohort trials: P=0.989; manipulation cohort delay trials vs control cohort trials: p=4.196 × 10^{-4}. (**C**) SWR rate for the pre-reward period for the subset of trials included in G, showing that even in neurofeedback trials with equal or higher velocities than delay or control trials, the SWR rate remains significantly elevated. Trial n are the same as in (**G**); manipulation cohort ranksum comparisons between NF and delay trials: p=3.923 × 10^{-11}, 1.155x10^{-8}, 1.902x10^{-18}, and 5.081x10^{-6}. Inset: Groupwise comparisons. Manipulation cohort NF trials vs control cohort trials: p=1.019 × 10^{-6}; manipulation cohort delay trials vs control cohort trials: p=7.958 × 10^{-3}. For all panels, ranksum p-values are corrected using the Benjamini-Hochberg method and all groupwise comparisons are performed using linear mixed effects models.

affect SWR rates, we selected the quartile of neurofeedback trials with the highest mean speeds and compared them to the quartiles of delay and control cohort trials with the lowest mean speeds. Even with much higher mean speeds (*Figure 3B*), the neurofeedback trials showed higher SWR rates than control and delay trials (*Figure 3C*). This analysis suggests that movement differences during the time spent at the center ports cannot explain the increased SWR rates observed prior to reward during neurofeedback trials.

The surprising stability of total SWR rate and count at both center ports across the two cohorts, despite changes in when those SWRs occurred relative to reward delivery, suggests that some regulatory mechanisms keep SWR generation within a set range for the center port phase of this task. We therefore asked whether we could detect evidence for this regulation on a trial-by-trial basis. Strict regulation within a single trial would predict that, for instance, trials with many SWRs prior to reward would likely have fewer SWRs post-reward. However, we did not find evidence for such a relationship consistently: the correlation coefficients between pre-reward and post-reward SWR counts were positive in some subjects and negative in others, and always very small in magnitude, indicating a weak relationship (r=–0.084, 0.130. 0.118, and –0.061 and p=1.47 × 10^{-7}, 3.84x10^{-6}, 8.60x10^{-9}, and 6.01x10^{-4}, for each subject in the manipulation cohort, respectively). Similarly, we did not find consistent correlations between pre-and post-reward SWR rate nor dwell time. These results suggest that while compensatory regulation of SWR generation may be evident across hundreds of trials, high variability in SWR rate, count, and dwell time across trials may prevent consistent detection of this effect at the level of single trials.

## Neurofeedback training preserves replay content during SWRs

SWRs are thought to support memory storage as a result of the specific patterns of spiking activity (replay) seen during these events (*Gridchyn et al., 2020*). Thus, we wanted to assess whether our SWR manipulation also increased replay during the targeted pre-reward period. As hippocampal CA1 neurons often fire reliably at particular locations in space ('place cells') while an animal moves through an environment, replay content has most often been studied with regard to its spatial content. Specifically, spatial content can be assessed by using a model that relates neural firing to specific locations in space. Inverting this 'encoding model' yields a 'decoding model' that can predict the spatial locations represented by the neural spiking within SWRs (*Zhang et al., 1998*).

We used a clusterless state-space model (*Chen et al., 2012*; *Deng et al., 2016*; *Denovellis et al., 2021*; *Kloosterman et al., 2014*) identical to the one we used previously to characterize the spatial content of replay for the animals in the control cohort (*Gillespie et al., 2021*). Briefly, we first project the 2D maze environment to 1D (*Figure 4A*). We next use times when the animal is moving through the maze to build a model that relates amplitude features of detected spikes to the location on the 1D maze where they were observed. We then can predict the spatial representation of spikes observed during movement (*Figure 4B*) and during SWRs (see Methods).

As expected, we observed a wide range of spatial representations during SWRs (*Figure 4C*). We define replay events as SWRs which predominantly contain spatial representations that are consistent with how a subject can experience the environment: those which represent a single location or a continuous trajectory through space (see Methods). These events frequently represent the subject's current location ('local' replay; *Figure 4C*, left two events) as well as locations or trajectories that encompass distant regions of the maze environment ('remote' replay; *Figure 4C*, middle four events). In small subset of cases, SWRs contain a mixture of disjoint spatial representations with no single location clearly dominant (fragmented events; *Figure 4C*, right), which do not meet our replay criteria and are excluded from analysis.

Across a variety of measures, we found that neurofeedback training preserved replay content. In SWRs during the pre-reward period, there was no difference in the fraction of SWRs that qualified as replay between the two cohorts, and within each manipulation cohort subject, there was no consistent difference between neurofeedback and delay trials (*Figure 4D*).

Replay of remote locations has been proposed to play a particularly important role in updating or maintaining memory in the absence of direct experience (*Gillespie et al., 2021*; *Gupta et al., 2010*; *Ólafsdóttir et al., 2015*). Consistent with our SWR findings, we find much higher rates of remote replay during the pre-reward period in neurofeedback trials compared to delay trials within the manipulation cohort, and low rates of remote replay during this time in trials from the control cohort (*Figure 4E*). However, if we consider both the pre-and post-reward periods combined, we find no difference in remote replay rate between the two cohorts for either trial type (*Figure 4F*). Within the manipulation cohort, neurofeedback trials have slightly but significantly higher rates of remote replay than delay trials over the combined pre- and post-reward period.

Prior work demonstrated that replay events in this task were enriched for specific, relevant past experiences that change as a function of goal location (*Gillespie et al., 2021*), and we saw the same biases in replay content in the manipulation cohort. We classified remote replay events based on the behavioral significance of the maze arm they represented and used a generalized linear model to assess how behavioral categories influence the likelihood of replay of a given arm (see Methods). As previously seen in the control cohort, we observed consistently enhanced representation of previous goal locations in replay events during the pre-reward period in both neurofeedback and delay trials within the manipulation cohort (*Figure 4G*). Similarly, we replicated the reduced prevalence of replay representing the immediately previously visited arm and a slightly increased prevalence of replay of the arm associated with the upcoming choice in both trial types of the manipulation cohort. These results also remain unchanged when we analyzed replay events during both pre- and post-reward time at the center ports (*Figure 4—figure supplement 1*).

Together, these results indicate that while the timing of replay is altered by the neurofeedback training, the content and behavioral relevance is preserved.

## Neurofeedback does not alter behavioral performance in a trial-by-trial manner

Finally, we wanted to assess whether neurofeedback training would have an impact on the performance of the spatial memory task. Since SWRs and remote replay rate were generally indistinguishable between the manipulation and control cohorts when the total time at the center ports was considered, we did not expect to find a large difference in behavior between the two cohorts. Within the manipulation cohort, however, neurofeedback trials generally showed higher SWR and remote replay rates, providing an opportunity to measure any behavioral consequences of the neurofeedback training. Our previous findings (*Gillespie et al., 2021*), suggested that SWRs and replay do not drive trial-by-trial choice behavior in this task, so we did not predict that the increase in SWRs on individual neurofeedback trials would necessarily improve performance on those trials. On the

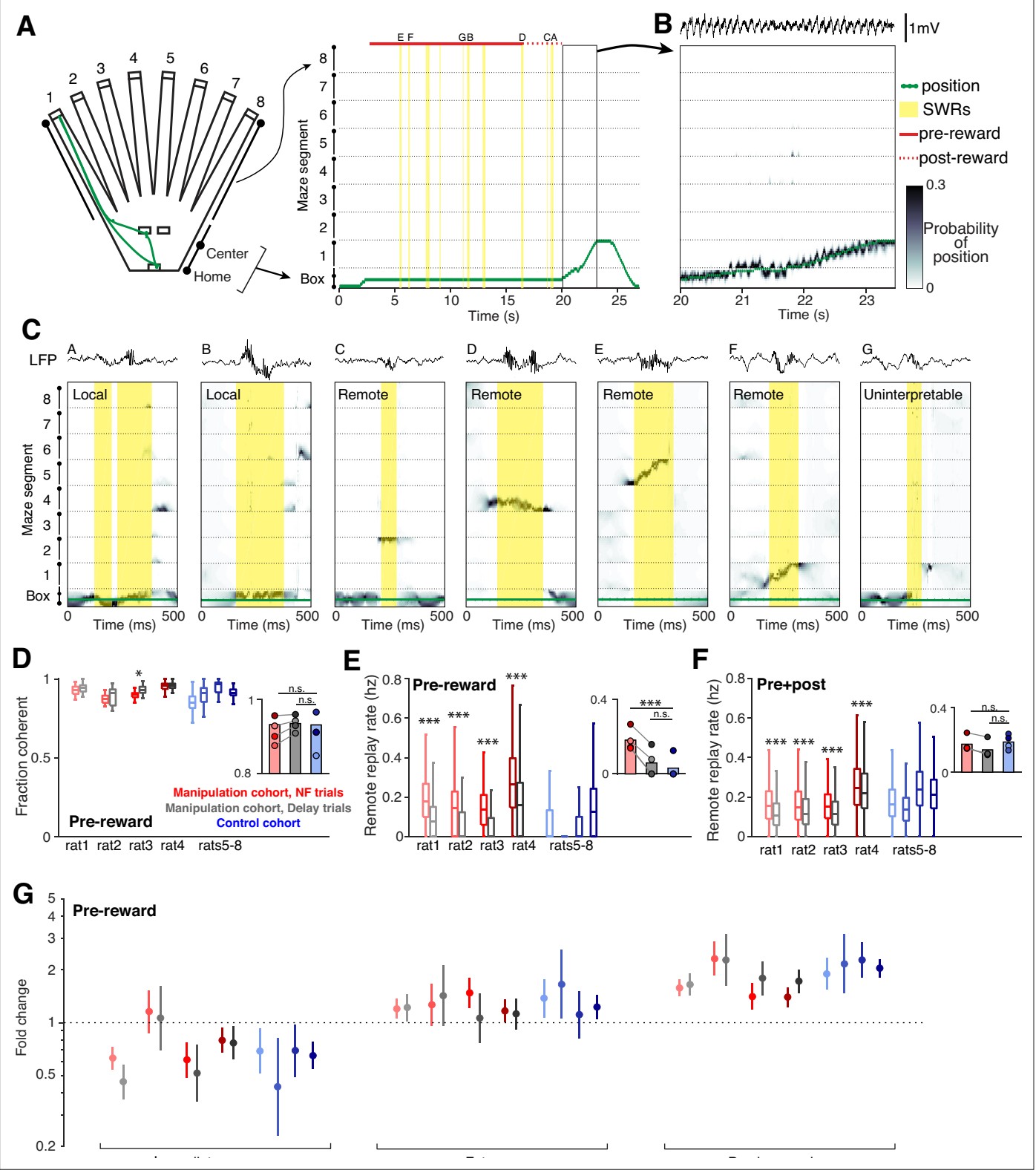

**Figure 4.** Neurofeedback preserves replay content. (**A**) The 2D maze is linearized to 1D for decoding efficiency; movement trajectory for an example neurofeedback (NF) trial is shown in green. Times of SWR events are highlighted in yellow, and the small letters indicate the SWRs that are shown in (**C**). (**B**) Decoding during movement times shows a close correspondence between decoded position and the subject's real position. (**C**) Several examples of decoding during SWRs, drawn from the example trial above. These include local events (two left), remote events (four middle), and one uninterpretable event (right). (**D**) The fraction of all SWRs per behavioral epoch that contain interpretable spatial content during pre-reward period at center ports.

*Figure 4 continued on next page*

*Figure 4 continued*

Manipulation cohort n=37, 14, 18, and 26 behavioral epochs per subject for each trial type and control cohort n=24, 32, 23, and 33 behavioral epochs. Manipulation cohort ranksum comparisons between NF and delay trials: p=0.1813, 0.5978, 0.0273, and 0.6738. Inset: Groupwise comparisons. Manipulation cohort NF trials vs control cohort trials: p=0.5523; manipulation cohort delay trials vs control cohort trials: p=0.2098. (**E**) Rate of remote replay events during pre-reward period. Manipulation cohort n=1843, 558, 1011, and 1513 NF trials and 1982, 535, 1038, and 1447 delay trials; control cohort n=2058, 2509, 1879, and 2795 trials per subject, respectively. Manipulation cohort ranksum comparisons between NF and delay trials: p=6.507 × $10^{-126}$, 9.703x$10^{-30}$, 2.991x$10^{-79}$, and 6.293x$10^{-55}$, respectively. Inset: Groupwise comparisons. Manipulation cohort NF trials vs control cohort trials: p=9.073 × $10^{-4}$; manipulation cohort delay trials vs control cohort trials: p=0.4304. (**F**) Rate of remote replay events during the pre- and post-reward periods combined. Trial n are the same as in (**E**). Manipulation cohort ranksum comparisons between NF and delay trials: p=7.666 × $10^{-47}$, 1.573x$10^{-6}$, 1.044x$10^{-16}$, and 5.768x$10^{-6}$. Inset: Groupwise comparisons. Manipulation cohort NF trials vs control cohort trials: p=0.6644; manipulation cohort delay trials vs control cohort trials: p=0.1477. (**G**) Generalized linear model coefficients quantify the extent to while replay of an arm is modulated by its behavioral relevance. Manipulation cohort n=1661, 392, 866, and 1281 NF trials and 1705, 367, 894, and 1213 delay trials per subject; control cohort n=1458, 1636, 1464, and 2181 trials, respectively. For panels D-F, all ranksum p-values are corrected using the Benjamini-Hochberg method and all groupwise comparisons are performed using linear mixed effects models (see Methods).

The online version of this article includes the following figure supplement(s) for figure 4:

**Figure supplement 1.** Replay content is consistent when considering all replay at the center ports.

other hand, we speculated that the requirement to generate SWRs could serve as a distracting or cognitively demanding additional aspect of neurofeedback trials that could impair subsequent arm choice on those trials, and if so, this would be an important factor to consider when designing future implementations of the neurofeedback paradigm.

We used several metrics to quantify task performance. First, during trials when the subject had not yet discovered a new goal arm ('search' trials), optimal behavior dictates that the animal should sample each arm without repeating visits to unrewarded arms until the goal arm is found. To measure performance on these trials, we quantified search efficiency: the fraction of search trials when the subject visits an arm not yet sampled. The manipulation cohort subjects show a slight improvement in search efficiency compared to control subjects, which only reached significance on delay trials (*Figure 5A*). However, rather than relating to the neurofeedback training, this difference could have been driven by the additional days of experience that the manipulation cohort has on the task, since the control cohort showed a subtle but consistent increase in this performance measure over time (*Figure 5—figure supplement 1*). Consistent with this possibility, we find no significant difference in this metric between neurofeedback and delay trials. We next asked whether 'redundant' search trials – when the subject chooses to sample an arm it has already visited – tended to be neurofeedback or delay trials. We found no bias toward either of the two trial types: neurofeedback trials accounted for approximately half of redundant search trials for each subject (*Figure 5B*), indicating that neither the neurofeedback nor delay trial type was consistently overrepresented among the trials with suboptimal behavior.

To quantify performance during trials when the goal arm had already been discovered by the subject and should be chosen again ('repeat' trials), we measured the fraction of subsequent trials during which the subject correctly visited the goal arm. We also saw no difference between the two cohorts or between neurofeedback and delay trials on this measure (*Figure 5C*). Within the manipulation cohort, we also found that error trials, when subjects do not choose to visit a known goal arm, were equally distributed between neurofeedback and delay trials (*Figure 5D*). Overall, these results suggest that on a single-trial timescale, the neurofeedback manipulation had neither a beneficial nor detrimental effect on behavior, in line with our previous findings (*Gillespie et al., 2021*).

## Discussion

We demonstrate that neurofeedback can be used to substantially alter the occurrence of SWRs and replay in rat subjects. Compared to a control cohort receiving no manipulation, subjects that experienced several weeks of neurofeedback training developed striking changes in when SWRs tended to occur. These subjects learned to greatly increase the rate of SWRs prior to reward delivery at the center ports, a period when subjects in the control cohort tended to have very few SWRs. This effect was strongest on the trials when SWRs were required (neurofeedback trials), and weaker but still evident on delay trials, when they were not. Importantly, we found that the neurofeedback training preserved behaviorally relevant, interpretable spatial replay during SWRs, and resulted in an increased

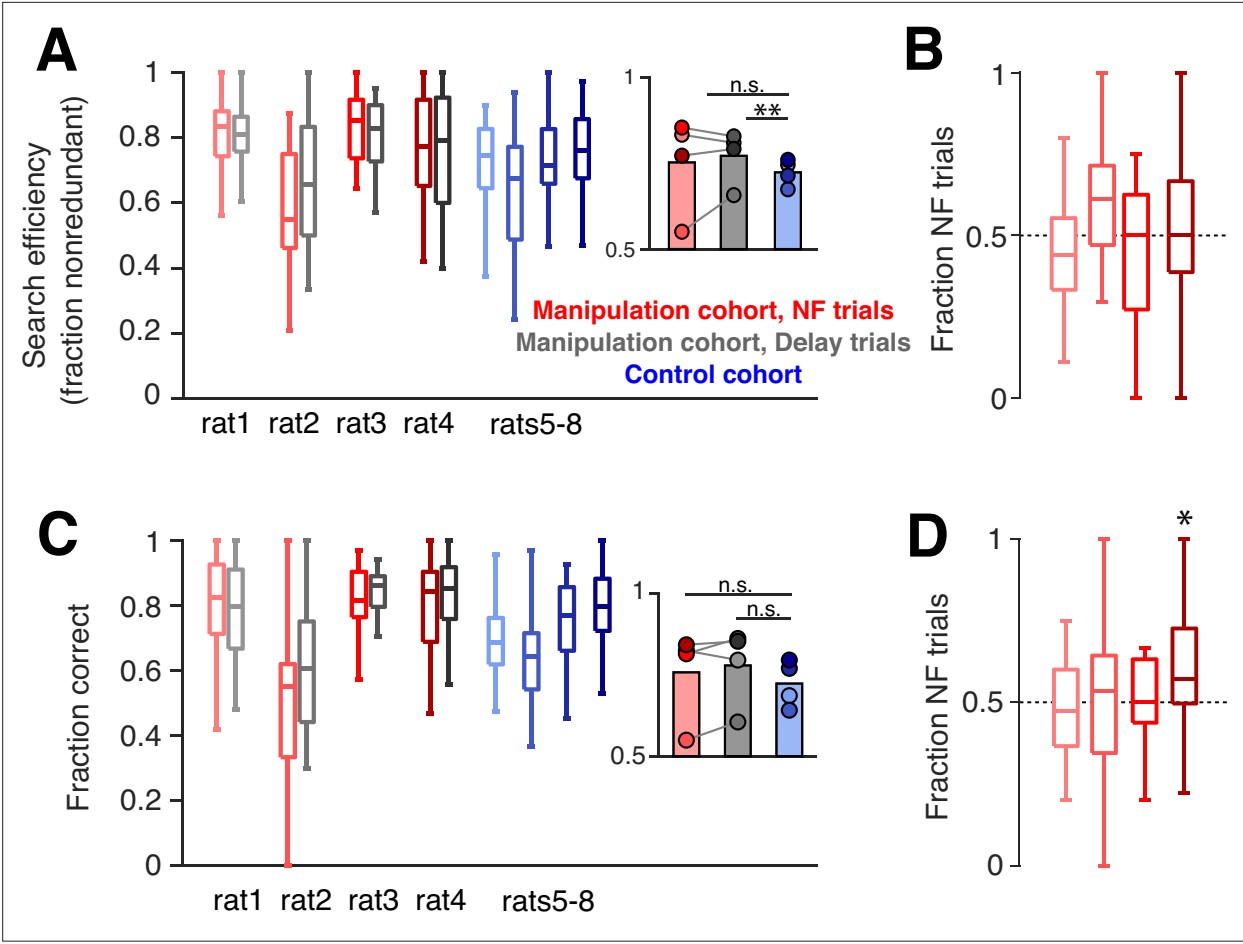

**Figure 5.** Neurofeedback does not alter behavioral performance. (**A**) During trials when the subject is searching for a new goal location, we quantify the fraction of trials in which the subject chooses an arm that has not yet been sampled. Manipulation cohort ranksum comparisons between neurofeedback (NF) and delay trials: p=0.9562, 0.7509, 0.7509, and 0.9562, respectively. Inset: Groupwise comparisons. Manipulation cohort NF trials vs control cohort trials: p=0.5425; manipulation cohort delay trials vs control cohort trials: p=0.0069. (**B**) The fraction of redundant search trials per epoch which are NF (vs delay). Two-sided sign test vs 0.5: p=0.1102, 0.2668, 1, and 1, respectively. (**C**) During trials when the subject has discovered the goal arm, the fraction of subsequent trials in which the goal arm is visited ('correct' choice). Manipulation cohort ranksum comparisons between NF and delay trials: p=0.7202, 0.7202, 0.8123, and 0.7202. Inset: Groupwise comparisons. Manipulation cohort NF trials vs control cohort trials: p=0.6946; manipulation cohort delay trials vs control cohort trials: p=0.2857. (**D**) The fraction of error repeat trials which are NF trials. Two-sided sign test: p=0.3770, 0.7744, 1, and 0.0227, respectively. For all panels, manipulation cohort n=37, 14, 18, and 26 behavioral epochs per subject, respectively, and control cohort n=24, 32, 23, and 33 behavioral epochs. For (A) and (C), all ranksum p-values are corrected using the Benjamini-Hochberg method and all groupwise comparisons are performed using linear mixed effects models.

The online version of this article includes the following figure supplement(s) for figure 5:

**Figure supplement 1.** Search efficiency increases with experience.

rate of remote replay on neurofeedback trials compared to delay trials. These results indicate that rapid neurofeedback is an effective approach to modulate physiologically relevant, memory-related patterns of activity in the hippocampus.

These results substantially extend a prior demonstration of operant conditioning of SWRs (*Ishikawa et al., 2014*) in three important ways. First, we used external feedback (food reward) as the reinforcer instead of intracranial stimulation, a change which makes the paradigm more amenable to clinical translation. Second, we administer the training in the context of a spatial memory task, which allows us to show that the neurofeedback training itself does not impose a cognitive burden. Finally, we show that the SWR events observed in the neurofeedback condition contain interpretable, behaviorally relevant spatial replay, suggesting that they are likely to contribute to normal replay function. Our findings also extend existing neurofeedback approaches by modulating the occurrence

of discrete memory-related events in the hippocampus, in contrast to longer-timescale modulation of continuous oscillations (*Engelhard et al., 2013*; *Klink et al., 2021*; *Reiner et al., 2014*; *Wang and Hsieh, 2013*).

Additionally, our neurofeedback paradigm extends existing approaches for positive manipulations of SWRs. First, we show that SWRs during the targeted interval contain robust and diverse replay that represents experiences relevant to the current state of the behavioral task, indicating that our method of SWR promotion recruits physiologically relevant ensembles during SWRs, does not overly prescribe which ensembles participate in SWRs, and allows replay content to evolve and adapt to changing task demands. Further, because subjects learn to modulate SWR rate, rather than simply generating a single suprathreshold event on command, it is likely that they learn to engage an SWR-permissive state during the targeted interval in which brain-wide neural activity and neuromodulatory tone also enter a SWR-permissive realm. This increases the likelihood that manipulated events could recruit downstream brain regions in physiologically relevant ways. Gaining experimental control over this SWR-permissive state offers compelling future opportunities to dissect extrahippocampal neural activity patterns and neuromodulatory influences on SWR generation.

Our results also revealed evidence for unexpected conservation of SWR rate and counts over the total time spent at the center ports. Despite the dramatic increase in SWR rate prior to reward delivery, we found that the overall number of SWRs produced was not different between the manipulation and control cohorts when the post-reward period was also taken into consideration. One can imagine that the existence of compensatory, potentially homeostatic, regulation of SWRs could be critical for adaptive behavior: an excess of replay could lead to intrusive, rigid memory or loss of behavioral flexibility, while too little could lead to forgetting or memory interference. Such possibilities are supported by evidence for important synaptic scaling properties of SWRs (*Norimoto et al., 2018*) and a prior finding that electrical disruption of SWRs during sleep caused increased SWR occurrence over longer timescales (*Girardeau et al., 2014*). Our results show that this regulation occurs during waking as well as during sleep, and further suggest that it occurs on short timescales, such as between pre- and post-reward periods at the center port in the current task. Moreover, our findings identify evidence for multiple systems that could interact to regulate SWR occurrence: in addition to learning to modulate SWR rate directly, subjects also learned to use behavioral strategies, such as changing the amount of time spent at a port, that had the effect of normalizing SWR counts.

In the context of a potential therapeutic manipulation, our study provides both a cautionary result and cause for optimism. We did not observe an improvement in behavioral performance due to the neurofeedback manipulation. This is consistent with prior results showing a decoupling between replay and trial-by-trial decisions in this task (*Gillespie et al., 2021*) and others (*Carey et al., 2019*); we predict that longer timescale manipulations of SWR rate (such as moving to blocks of neurofeedback vs delay trials rather than randomly interleaving the two trial types) would affect memory consolidation and would show behavioral impacts. We also observed evidence for compensatory changes in the overall number of SWRs and replay events between the two cohorts prior to the memory-guided decision phase of the trial (outer arm choice). This suggests that in our young and healthy subjects, various homeostatic mechanisms act to maintain SWRs and the associated replay events at advantageous levels, potentially limiting the extent to which we are able to shift SWR rates beyond normal levels.

At the same time, our study demonstrates that it *is* possible to use behavioral feedback to modulate the rates of SWRs and associated replay events during a specific time period. In subjects with compromised SWRs, such as rodent models of Alzheimer's disease, even a moderate increase in SWRs or replay could move the system back towards healthy levels and potentially improve memory performance. Since the neurofeedback paradigm depends on the occurrence of at least a low endogenous rate of SWR occurrence, it would be important to implement neurofeedback training as a relatively early interventional strategy prior to extensive neurodegeneration, and training may take longer in aged or impaired subjects. Our results also raise the possibility that reduced SWR rates in disease models could be caused by a failure of homeostatic regulatory mechanisms. If this is the case, then a neurofeedback approach may even more effectively promote SWRs in the disease case, because it will not have to overcome internal regulation. Our current findings therefore serve as useful 'proof-of-concept' results that can inform and shape the implementation and optimization of SWR-focused neurofeedback interventions in disease models. Importantly, when combined with minimally

invasive techniques for detecting SWRs and replay in humans, such as MEG (*Liu et al., 2019*; *Nour et al., 2021*), this technique could be further adapted to patient populations to assess whether SWR-triggered neurofeedback in humans could lead to similar enhancement or restructuring of replay, and ultimately whether such an intervention could be therapeutic.

# Methods

## Resource availability

### Lead Contact

Requests for further information, resources, or code should be directed to and will be fulfilled by the lead contact, Anna Gillespie (annagill@uw.edu).

### Materials availability

No new reagents were generated by this study.

## Experimental model and subject details

Data from four male Long-Evans rats (Charles River RRID:RGD_2308852; aged 4–9 months; 450–550 g) comprise the neurofeedback cohort of animals included in this study. We also include data from a cohort of four additional male Long Evans rats used for a previous study who did not receive neuro-feedback training (*Gillespie et al., 2021*). All animals were kept on a 12 hr light-dark cycle (lights on 6am-6pm) with ad libitum access to food (standard rat chow) in a temperature- and humidity-controlled facility. Beginning 1 week prior to and extending throughout behavioral training and data acquisition, rats were singly housed and food restricted to a maximum of 85% of free-feeding weight. All procedures were approved by the Institutional Animal Care and Use Committee at the University of California, San Francisco (Protocol # AN191130).

## Method details

### Behavioral training

Food-restricted rats first complete two 40 min sessions of training on a linear track to receive liquid food reward (evaporated milk +5% sucrose) from automated, photogated reward ports at each end of the track. Motivated subjects (>50 rewards on final session) advance to learn a complex spatial memory task. This task takes place on a large, walled maze equipped with reward ports, the same as used in *Gillespie et al., 2021*. Pretraining occurs as described in *Gillespie et al., 2021*; briefly, subjects are gradually taught to reliably perform the task, with each trial consisting of an initiation poke at the home port and small (50 µL) reward delivery, then an extended dwell period at whichever of the two central ports illuminates, and finally a visit to one of the arm ports. During pretraining, both center ports require the rat to keep its nose in the port for a delay period, randomly chosen from a range, up to 5–10 s. After the delay is complete, a tone plays and a small reward (50 µL) is delivered. Finally, the subject must choose one arm to visit; only one out of the eight arms will deliver reward at any given time, so the rat must visit different arms over several trials to discover the goal arm. The rat must remember this location and return to it on subsequent trials to receive a large reward (150 µL) at the arm port. After a certain number of rewards earned at the goal arm (10-15), the goal arm will change, causing the rat to need to search among arms to find the new goal location. Later in training, goal locations are changed after 4–12 repeat visits. Any deviation from the trial structure described results in a 30–45 s 'timeout' during which all ports deilluminate; no reward will be delivered from any port until the timeout period is up and the subject initiates a new trial at the home port. Pretraining continues until the subject can perform reliably; usually 14 days of training with several 30–60 min training epochs per day.

### Neural implant

Best performing subjects are selected for surgery and allowed ad libitum food access without behavioral training for at least 5 days prior to implant. Each implant houses 30 independently moveable nichrome tetrodes (12.7 um, California Fine Wire), cut at an angle and gold plated to a final impedance of 210–350 kOhms. The implant is stereotactically positioned such that each of the bilateral

cannulae containing 15 tetrodes is centered at 3.6–4 mm AP, ± 2.6 mm ML relative to skull bregma. A screw placed over cerebellum serves as global reference. Subjects were allowed at least 5 and up to 10 days for recovery after surgery, with ad libitum food and daily tetrode lowering until dorsal CA1 cell layer is reached. One tetrode per hemisphere is targeted to the corpus callosum to serve as a local reference for each cannula.

## Online SWR detection and neurofeedback training

After recovery, manipulation cohort subjects are allowed 1–2 days of the pretraining task paradigm to resume reliable performance before beginning the neurofeedback phase of training. One center port becomes the neurofeedback port, at which subjects must hold their nose in the port until a suprathreshold SWR is detected in order to trigger a sound cue and reward. The other port (delay port) also requires subjects to hold their nose in the port, but the sound cue and reward is delivered after a certain amount of time, independent of neural activity. This approach constrains the behavior of the animal to be overtly similar during both trial types. Delay times are chosen from the latencies until trigger SWR generation from the 8 prior neurofeedback trials, which results in the times spent at neurofeedback and delay ports being approximately yoked over the course of the epoch. We did not use the more traditional approach used in SWR intervention studies in which the manipulation is performed at a fixed offset after SWR detection (*Fernández-Ruiz et al., 2019*; *Girardeau et al., 2009*; *Jadhav et al., 2012*), because in our paradigm this would introduce a systematic difference in the time spent waiting for reward at the two ports and would thus risk biasing the subject's behavior. The neurofeedback/delay port locations were counterbalanced across subjects. Subjects complete two to three 30–90 min behavioral epochs each day; 30-min rest epochs in a small rest box precede and follow each behavioral epoch.

During behavioral epochs, SWRs are detected online by monitoring the smoothed envelope of ripple filtered (100–400 Hz) LFP from 4 to 6 CA1 cell layer tetrodes (*Figure 1—figure supplement 1A*). The mean $\mu_{est}$ and standard deviation $\sigma_{est}$ of the absolute value of the ripple filtered trace for each tetrode is calculated using an iterative procedure *Jadhav et al., 2012*:

$$\mu_{est}(n) = \mu_{est}(n-1) \cdot (N_{smooth} - 1)/N_{smooth} + |x|/N_{smooth}$$

$$\sigma_{est}(n) = \left| |x| - \mu_{est}(n-1) \right| - \sigma_{est}(n-1)/N_{smooth} + \sigma_{est}(n-1)$$

Here, x is the absolute value of the ripple filtered LFP, sampled at 1500 Hz, and $N_{smooth}$ is the number of samples used for smoothing (10,000). Values are measured prior to each behavioral epoch while the subject is alert and active in the rest box, allowed to stabilize, fixed, and used to determine the SWR detection threshold for the subsequent behavioral epoch. During behavior, the envelope $v_{est}$ of the ripple filtered LFP from the monitored tetrodes is calculated and smoothed using an asymmetrical iterative estimator that allows for rapid detection of power increases *Jadhav et al., 2012*:

$$v_{est}(n) = v_{est}(n-1) + g_i(n-1) \cdot |x| - v_{est}(n-1)$$

where $g_i$ is 0.2 when the envelope was decreasing ($|x| \leq v_{est}(n-1)$) and $g_i$ is a moving average of 1.2 and the last 19 values of $g_i$ when the envelope was increasing ($|x| > v_{est}(n-1)$). SWR detection is triggered when the ripple envelope of 2 or more tetrodes crosses the detection threshold simultaneously. Trigger detection is followed by a delay of 50–100ms before the sound cue and reward is delivered (75ms for rat1, 100ms for rat2, 50ms for rat3 and rat4; see *Figure 1—figure supplement 1B*), in order to minimize the chance of aborting or interrupting the ongoing SWR event with abrupt stimuli. The set of detection tetrodes changes each day and always includes tetrodes from each hemisphere. The threshold for trigger SWR detection is gradually raised, across 14–20 days, from 4 sd to 16–20 sd, and then is maintained at the maximum threshold for 7–10 days (*Figure 1—figure supplement 1E*). Within each behavioral epoch, the detection threshold starts low (4 sd) and is raised to the maximum threshold for that day over the course of the first ~10 neurofeedback trials.

## Data collection and processing

Continuous 30 kHz neural data, environmental events (port beam breaks, lights, reward delivery, online SWR triggers, etc), online position tracking of a head-mounted LED array, and 30 Hz overhead video were collected and synchronized using a SpikeGadgets data acquisition system. LFP was generated

by filtering between 0.1 and 300 Hz and was referenced to a tetrode located in ipsilateral corpus callosum. In parallel, the continuous signal was referenced and filtered between 600 and 6000 Hz, and spike events were detected when the voltage exceeded 100 µV on any channel of a tetrode.

### Offline SWR detection

SWRs were detected offline using a consensus method incorporating information from all CA1 cell layer tetrodes (*Figure 1—figure supplement 1A*). LFP traces were filtered for ripple band activity (150–250 Hz), squared, summed over tetrodes, smoothed with a 4ms sd gaussian kernel, and the square root was taken (*Kay et al., 2016*). The resultant consensus trace was used for event detection with a threshold set 2 sd above the behavioral epoch mean and a minimum suprathreshold duration of 15ms. SWR start time was defined as the time when the consensus trace crossed epoch baseline prior to threshold crossing and end time was when the consensus trace returned to baseline (*Gillespie et al., 2021*). SWR size was calculated as the maximum threshold value at which the event would have been detected (in units of sd), and SWR length was calculated as the difference between start and end times for each event.

### Histology

At the conclusion of data collection, tetrode locations were marked with electrolytic lesions and the subject was transcardially perfused with paraformaldehyde 24 hr later. The brain was fixed, sectioned into 50 µm slices, and Nissl stained to enable the localization of tetrode tips (*Sosa et al., 2020*).

## Quantification and statistical analysis

All analyses were performed using custom code written in Matlab 2020a (Mathworks) and Python 3.6.

### Statistical tests

All quantification of manipulation cohort trials (with the exception of *Figure 2—figure supplement 1A,B*) use only trials from the stable period of neurofeedback training. For the control cohort, trials from all days of reliable post-implant task performance are used. Groupwise comparisons are performed using generalized mixed-effects models to account for and measure both individual variability and group effects. Using the Matlab *fitglme* function, we construct a model in which each sample (SWR event features, trials, etc) is assigned to an individual subject as well as to a cohort (manipulation or control). Linear models are used for rate, dwell time, SWR amplitude, and proportion measures, while a Poisson distribution is used for count measures. Each model is designed to compare samples from all trials of the control group to samples from neurofeedback and delay trials from the neurofeedback cohort for a specific time period (for instance, pre-reward-delivery at the center ports). To compare neurofeedback and delay trials within each manipulation cohort subject, we use nonparametric ranksum tests. We then perform post-hoc correction for multiple comparisons using the Benjamini-Hochberg method to control false discovery rate (*Groppe, 2022*).

### Neurofeedback wait time prediction

For each behavioral session during the first ~7 days of pretraining and during early stages of neurofeedback training, SWRs occurring during center port times (pre-and post- reward) were binned by size and occurrence rate was calculated by dividing the number of SWRs within a given bin by the total time spent at the center port, cumulative over all behavioral epochs. Occurrence rates were only calculated for size bins containing at least 10 events. For later days, a predicted wait time was generated for each neurofeedback trigger event based on the occurrence rate calculated for that size of event. Events without a valid occurrence rate size bin were excluded from analysis. For each subject, actual and predicted wait times at the neurofeedback port were concatenated over behavioral sessions and smoothed with a 200-trial moving average (*Figure 2—figure supplement 1B*).

### SWR size quantification

The structure of the neurofeedback phase of the task guarantees that each neurofeedback trial includes *exactly one* suprathreshold SWR event: the pre-reward period continues until the first suprathreshold event occurs and then immediately ends. However, no such criteria is enforced upon delay

trials. Therefore, comparing SWR size taking all events during the pre-reward period of each trial type results in a biased comparison. To control for this bias, the suprathreshold SWR at the end of each neurofeedback trial as well as any suprathreshold SWRs detected during delay trials are excluded from certain analyses (*Figure 2—figure supplement 1C, D, and E*).

### Timewise SWR rate quantification

Each trial, from initial center port nosepoke to exit from the center port, was divided into 0.5 s bins and the count of SWRs in each time bin was calculated. Trials were aligned to the tone/food reward delivery and the mean and S.E.M. for each time bin was calculated across all trials and converted to Hz. Only bins with at least 100 trials contributing data were included. The trigger SWR for each neurofeedback trial was excluded from analysis to avoid a large spike in the trace just prior to reward delivery that distracted from the comparisons between the SWR rates before and after the trigger event.

### Clusterless decoding

Marks were calculated for each event as the amplitude of the spike waveform on each of the four tetrode channels at the time of peak amplitude following threshold crossing. Bayesian marked point process clusterless decoding with a state space movement model was performed as described in *Denovellis et al., 2021* and exactly as implemented for this maze environment in *Gillespie et al., 2021*. Briefly, tracked position was linearized such that positions in the central 2D area of the maze were collapsed onto a single line extending from the home well to the edge of the arms. Linearized position was binned into 5 cm bins. The encoding model comprised of all marks from CA1 tetrodes collected when the subject was moving >4 cm/s, and 2ms time bins were used. Two movement models were included: one with an equal probability of moving one position bin forward, back, or staying in the same position (continuous state), and one with an equal probability of moving to any other possible position bin (fragmented state). A uniform discrete transition matrix governed transitions between movement states, and a resultant joint posterior was computed. For the decoding of movement times the spike being decoded was always excluded from the encoding model.

### Replay content analysis

Events were categorized as interpretable if the probability of the continuous movement model exceed a 0.8 threshold during the event and if at least 30% of the posterior density was in a single maze segment. Interpretable events were categorized as local if the majority of posterior density was located in the same maze segment as the animal's current position and remote otherwise. Poisson generalized linear models were used to quantify the effect of a given behavioral arm category on the likelihood of replay (*Gillespie et al., 2021*). Search and repeat trials from the second goal block onwards were used for GLM analysis (so that all trials included a valid previous goal option). Using the MATLAB *fitglm* function, we constructed one binary predictor per arm category (previous arm, future arm, and previous goal arm); each trial contributed one entry per maze arm with a one if the arm was described by the category and a zero if it was not. The response variable for each entry was the number of replays of the arm seen on that trial. The resultant model coefficients were converted to fold change by exponentiation and confidence intervals were calculated using MATLAB's *coefCI* function.

## Acknowledgements

The authors are grateful to Max Ladow and Rhino Nevers for their assistance with data collection, to Jennifer Guidera for her contributions to analysis methods, and to Tom Davidson, Demetris Roumis, P Dylan Rich, Jon Rueckemann, and all members of the Frank lab for insightful discussions throughout the project. We also thank Alan Kaplan and the Lawrence Livermore National Laboratory High Performance Computing Center for providing computational resources.

## Additional information

### Competing interests

Anna K Gillespie, Loren M Frank: U.S. Patent Application No. 62/887,875 on SWR modulation via neurofeedback has been filed by the University of California, San Francisco with AKG and LMF listed as inventors. The other authors declare that no competing interests exist.

### Funding

| Funder | Grant reference number | Author |
| --- | --- | --- |
| Howard Hughes Medical Institute | | Loren M Frank |
| Simons Foundation | 500639 | Anna K Gillespie |
| Simons Foundation | 521921 | Loren M Frank |
| BRAIN Initiative | K99AG068425 | Anna K Gillespie |

The funders had no role in study design, data collection and interpretation, or the decision to submit the work for publication.

### Author contributions

Anna K Gillespie, Conceptualization, Resources, Data curation, Software, Formal analysis, Supervision, Funding acquisition, Validation, Investigation, Visualization, Methodology, Writing – original draft, Project administration, Writing – review and editing; Daniela Astudillo Maya, Data curation, Investigation; Eric L Denovellis, Software, Methodology; Sachi Desse, Investigation; Loren M Frank, Conceptualization, Supervision, Funding acquisition, Writing – original draft, Project administration, Writing – review and editing

### Author ORCIDs

Anna K Gillespie ⓘ https://orcid.org/0000-0003-0980-2408
Eric L Denovellis ⓘ https://orcid.org/0000-0003-4606-087X
Loren M Frank ⓘ https://orcid.org/0000-0002-1752-5677

### Ethics

All procedures in this study were approved by the Institutional Animal Care and Use Committee at the University of California, San Francisco (Protocol # AN191130).

Reviewer #2 (Public review): https://doi.org/10.7554/eLife.90944.3.sa1
Reviewer #3 (Public review): https://doi.org/10.7554/eLife.90944.3.sa2
Author response https://doi.org/10.7554/eLife.90944.3.sa3

## Additional files

### Supplementary files

• MDAR checklist

### Data availability

All raw neurophysiology and behavioral data used for this study are publicly available on the DANDI Archive (https://dandiarchive.org/dandiset/000115 and https://dandiarchive.org/dandiset/000629). Processed data is available at https://doi.org/10.6084/m9.figshare.24919590.v1. All code used to analyze and visualize this processed data is available at https://github.com/LorenFrankLab/Gillespie_eLife_2024 (copy archived at *Gillespie, 2024*).

The following datasets were generated:

| Author(s) | Year | Dataset title | Dataset URL | Database and Identifier |
|---|---|---|---|---|
| Gillespie AK, Astudillo Maya DA, Denovellis EL, Desse S, Frank LM | 2024 | Neurofeedback training can modulate task-relevant memory replay rate in rats | https://dandiarchive.org/dandiset/000629 | DANDI, 000629 |
| Gillespie AK | 2024 | Processed data for analysis and visualization of results described in Gillespie et al, eLife 2024, "Neurofeedback training can modulate task-relevant replay rate in rats" | https://doi.org/10.6084/m9.figshare.24919590.v1 | figshare, 10.6084/m9.figshare.24919590.v1 |

The following previously published dataset was used:

| Author(s) | Year | Dataset title | Dataset URL | Database and Identifier |
|---|---|---|---|---|
| Gillespie AK, Astudillo Maya DA, Denovellis EL, Liu D, Kastner D, Coulter ME, Roumis DK, Frank LM | 2021 | Hippocampal replay reflects specific past experiences rather than a plan for subsequent choice | https://dandiarchive.org/dandiset/000115 | DANDI, 000115 |

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
