## [Editor Report · eLife assessment]

This study tests the effects of using neurofeedback, in the form of reward delivery when large sharp wave-ripples (SWRs) are detected, on neurophysiological and behavioral measures. The results are **important**, and the authors provide **convincing** evidence that the rate of SWRs increased prior to reward delivery and decreased in the period after reward delivery, with no significant effect on memory performance. The ability to manipulate SWR rate in a naturalistic way is an exciting new tool for studies that seek to understand the function of SWRs.

---

## [Referee Report · Reviewer #2 (Public review)]

Gillespie et al. introduced a novel neurofeedback (NF) procedure to train rats in enhancing their sharp-wave ripple (SWR) rate within a short duration, a key neural mechanism associated with memory consolidation. The training, embedded within a spatial memory task, spanned 20-30 days and utilized food rewards as positive reinforcement upon SWR detection. Rats were categorized into NF and control groups, with the NF group further divided into NF and delay trials for within-subject control. While single trial differences were elusive due to the variability of SWR occurrence, the study revealed that statistically rats in NF trials exhibited a notably higher SWR rate before receiving rewards compared to delay trials. This difference was even more pronounced when juxtaposed with rats not exposed to NF training (control group). The unique design of blending the NF phase with the memory dependent spatial task enabled the authors to analyze whether the NF training influence the task performance and replay content during SWRs across three different conditions (NF trials, delay trials and control group). Interestingly, despite the NF training, there was no significant improvement or decline in the performance of the spatial memory task, and the replay content remained consistent across all three conditions. Hence, the operant conditioning only amplified the SWR rate before reward in NF trials without altering the task performance and the replay content during SWR. Moreover, considering the post-reward period, the total SWR count was consistent across all conditions as well, meaning the NF training also do not affect the total SWR count. The study concludes with the hypothesis of a potential homeostatic mechanism governing the total SWR production in rats. This research significantly extends previous work by Ishikawa et al. (2014), offering insights into the NF training with external reward on the SWR rate/counts, replay content and task performance.

Strengths:

- Integration of NF task and spatial memory task in a single trial

The integration of NF training within a spatial memory task poses significant challenges. Gillespie and colleagues overcame this by seamlessly blending the NF task and the spatial memory task into a single trial. Each trial involved a rat undergoing three steps: First, initiating a trial. Second, moving to either the NF port or the delay trial port, as indicated by an LED, and then maintaining a nosepoke at one of the center ports. During this step, the rat had to keep its nose (in the NF port) until a sharp-wave ripple (SWR) exceeding a set threshold was detected, which then triggered a reward, or until a variable time elapsed (in the delay port). Third, the rat would choose one of eight arms to explore before starting the next trial. This integration of the two tasks (step two as the NF task and step three as the spatial memory task) facilitated a direct analysis of the impact of NF training on behaviorally relevant replay content during SWRs and the performance in the spatial memory task.

- Clear Group Separation

A robust study design necessitates clear distinctions between experimental conditions to ensure that observed differences can be attributed to the variable under investigation. This study meticulously categorized rats into three distinct conditions: NF trials, delay trials (for within-subject control), and a control group (for across-subject control). Furthermore, for each trial, the times of interest (TOI) were separated into pre-reward and post-reward periods. This clear separation ensures that any observed differences in SWR rates and other outcomes can be confidently attributed to the effects of neurofeedback training during specific time periods, minimizing potential confounding factors.

- Evidence of SWR rate modulation

The study's results offer compelling evidence that rats can be trained to modulate their SWR rates during the pre-reward period. This is evident from the observation that rats in the NF trials consistently displayed a higher SWR rate before receiving rewards compared to those in delay trials or the control group (Fig. 2). Such findings not only validate the efficacy of the NF paradigm but also underscore the potential of operant conditioning in influencing neural mechanisms. The observation that rats were able to produce larger SWR events by modulating their occurrence rate, rather than merely waiting for these events, suggests a learned strategy to generate them more efficiently.

- Evidence of SWR count homeostasis

A notable finding from the study was the observation of a consistent total SWR count during both pre-reward and post-reward periods across all conditions, despite the evident increase in SWR rates during the pre-reward period in NF trials. This points to a potential homeostatic mechanism governing SWR production in rats. This balance suggests that while NF training can modulate the timing and rate of SWRs over a short duration, it doesn't influence the overall count of SWRs over a longer period. Such a mechanism might be essential in ensuring that the brain neither overcompensates nor depletes its capacity for SWRs, maintaining the overall neural balance and functionality. This discovery deepens our understanding of neural mechanisms and highlights potential avenues for future research into the regulatory processes governing neural activity.

In this revision, the paper explores a neurofeedback technique in rats that modulates hippocampal sharp-wave ripple (SWR) rates, crucial for memory replay, without altering the content of those replays. The study demonstrates that neurofeedback can specifically increase SWR rates during a task's pre-reward phase. Revisions address concerns about movement's impact on SWR rates, clarify the statistical approach used, and modify the title for accuracy, now emphasizing the modulation of memory replay rates rather than suggesting alterations to memory content itself. I think all the concerns in the previous version have been addressed.

---

## [Referee Report · Reviewer #3 (Public review)]

Summary:

This study implements an innovative neurofeedback procedure in rats, providing food reward upon detection of a sharp wave-ripple event (SWR) in the hippocampus. The elegant experimental design enables a within-animal comparison of the effects of this neurofeedback procedure as compared to a control condition in which equivalent reward is provided in a non-contingent manner. The neurofeedback procedure was found to increase SWR rate, followed by a compensatory reduction in SWR rate. These changes in SWR rate were not accompanied by any changes in memory performance on the memory-guided task.

Strengths:

The scientific premise for the study is outstanding. It addresses an issue of high importance, of developing ways to not merely describe correlations between SWRs (and their content) and memory performance, but to manipulate them. The authors argue clearly and convincingly that even studies that have performed causal manipulations of SWRs have important confounds and limitations, and most importantly for translational purposes, they are all invasive. So, the idea of developing a potentially non-invasive neurofeedback procedure for modulating SWRs is compelling both as an innovative new experimental manipulation in studies of SWRs, and as a potentially impactful therapeutic avenue.

In addition to addressing an important issue with an innovative approach, the study has many other strengths. The data unambiguously show that the method is effective at increasing SWR rate in each individual subject. The experimental design allows within-subject comparison of neurofeedback and control trials, where the subjects wait an equivalent amount of time. The careful analyses of SWR properties and their content establish that neurofeedback SWRs are comparable to control SWRs. The data add further evidence to the notion that SWR rate is subject to homeostatic control. The paper is also exceptionally well written, and was a pleasure to read. So, there is a clear technical advance, in that there is now a method for increasing SWR rate non-invasively, which is rigorously established and characterized.

Weaknesses:

The one overall limitation I find with this study is that it is unclear to what extent the same (or better) results could have been obtained using behavior-feedback instead of neuro-feedback. Because SWR rates are generally higher during states of quiescence compared to active movement or task engagement, it is possible that reinforcing behaviorally detected quiescent states (e.g. low movement) would indirectly increase SWR rates. The authors include an important control analysis showing higher SWR rates in the neurofeedback condition even when movement speed is controlled for by subsetting the data, demonstrating that changes in movement speed cannot be the only explanation of the results. At the same time, the observation that all 4 subjects had lower movement speeds during neurofeedback compared to control trials suggests that neurofeedback is likely reinforcing both overt (behavior) and covert (SWR) processes. Understanding the relative contributions of each to the observed SWR increase would help clarify whether the neurofeedback approach is worth the additional effort and expense compared to behavioral feedback.

---

## [Author Response]

The following is the authors’ response to the original reviews.

Major change:

All three of our reviewers raised the possibility that changes in movement during the time spent at the center ports could have contributed to changes in SWR rates. Analyses to address this possibility, based on the examination of trials with high and low speeds, were originally included in the supplement but we did not sufficiently highlight and explain these results. To rectify this, we have moved these results into a new main Figure 3 and now include a paragraph describing our interpretation of these results (page 9). We also include a more detailed description of the subjects’ behavior during port times – namely, that all subjects must remain quite stationary while at the reward ports in order to keep their nose in a specific position which keeps the port triggered. As a result, all subjects maintain head speeds well below our typical speed threshold for immobility while at the ports. This leads us to predict that any feedback based on periods of immobility alone (as requested by Reviewer 3) would show results very similar to our Control cohort and would not alter SWR rates seen during neurofeedback trials.

Minor changes:

(1) Reviewer 1 observed our that reported statistics appeared to be missing an interaction term showing that neurofeedback differentially affected the SWR rate/count pre- and postreward. We apologize for a lack of clarity here: we fit pre- and post-reward times with separate linear mixed effects models, so this interaction term is neither expected nor defined in our model. We have added a sentence clarifying this aspect of our LME approach in the Methods section: “Each model is designed to compare samples from all trials of the control group to samples from neurofeedback and delay trials from the neurofeedback cohort for a specific time period (for instance, pre-reward-delivery at the center ports).” Combining both times in the same model would require adding an additional hierarchical level in order to preserve the pairing of the pre- and post-reward time period for each trial, which we are concerned would complicate the formulation and interpretation of the model. However, the reviewer raises a good point that the comparison between these two time periods reveals an additional difference between the trial types: SWR rate remains relatively consistent between the pre- and post-reward periods during neurofeedback trials, while delay and control trials show a clear increase in SWR rate between the two time periods. To visualize and quantify this effect, we calculated the difference in SWR rates between the two time periods and now include this plot as Supplementary Figure 2F, which is referenced in page 8 of the main text.

(2) Reviewer 2 found our original title, “Neurofeedback training can modulate task-relevant memory replay in rats” to be misleading and suggestive of a manipulation to memory content. We are in complete agreement with the Reviewer in that our manipulation does not alter replay content, so to be more specific and accurate, we have changed our title to their suggestion “Neurofeedback training can modulate task-relevant memory replay rate in rats” accordingly.

(3) Reviewer 2 also requested that we include analyses quantifying baseline SWR rates for each of our experimental subjects. Although we initially considered reporting our results in measures of change relative to each individual animal’s baseline, we decided against this approach for several reasons.

First, it is important to clarify that we extensively train the animals on the task prior to implant, so we do not have access to a truly naïve, pre-behavior baseline SWR rate for any of our subjects. However, because the pre-implant training is conducted consistently between our neurofeedback and our control cohort, we have no reason to believe that the behavioral training prior to implant would introduce differences in SWR rate between the cohorts. Indeed, we find no difference in post-reward SWR rate (or SWR rate at the home well) when we quantify the first 250 trials of post-implant behavior for each subject (see panel A below). Note that we cannot compare the pre-reward SWR rate at this point, because it is influenced by the task structure which guarantees at least one SWR in each neurofeedback trial pre-reward.

Further, we do find that SWR rate is quite consistent over many days of task performance in the control cohort (show for the post-reward period in panel B below). This suggests that comparing the post-neurofeedback training SWR rates for the neurofeedback cohort to SWR rates throughout the training for the control cohort is not likely to be confounded by differing amounts of training experience. This is supported by our analyses in Figure 2 which show no differences in SWR rate between the two cohorts when considering pre- and post-reward times combined.

**Author response image 1. sa3fig1:** (A) SWR rate calculated during the post-reward period at the center port for the first 250 trials of postimplant behavior for each animal. Trials of all types are included (ie both neurofeedback trials and delay trials for the manipulation cohort). Groupwise comparison p=0.192. (B) Mean SWR rate during the post-reward period at the center port for each behavioral training epoch shows no systematic change over time across subjects within the control cohort.

Finally, within each cohort, we found the overall SWR rates to be quite consistent across animals. If each subject in the neurofeedback cohort had shown dramatically different SWR rates at the beginning of neurofeedback training, we would have needed to express the effect of neurofeedback training relative to baseline for each animal. However, since the range of SWR rates were highly comparable, we felt that it was more accessible, and easier to place our results within the context of the literature, by expressing our results as simple SWR rates themselves rather than measures of relative change. Within the neurofeedback cohort, comparing neurofeedback to delay trials is inherently matched for baseline SWR rate since these comparisons are made within the same animal.

(4) Finally, Reviewer 2 raises the possibility that older animals or those with cognitive deficits might respond to neurofeedback differently. We entirely agree with this possibility, and note this in our Discussion section: “Since the neurofeedback paradigm depends on the occurrence of at least a low endogenous rate of SWR occurrence, it would be important to implement neurofeedback training as a relatively early interventional strategy prior to extensive neurodegeneration, and training may take longer in aged or impaired subjects.”